# The Role of a Lung Vascular Endothelium Enriched Gene TMEM100

**DOI:** 10.3390/biomedicines11030937

**Published:** 2023-03-17

**Authors:** Jiakai Pan, Bin Liu, Zhiyu Dai

**Affiliations:** 1Division of Pulmonary, Critical Care and Sleep, University of Arizona, Phoenix, AZ 85004, USA; 2Department of Internal Medicine, College of Medicine-Phoenix, University of Arizona, Phoenix, AZ 85004, USA; 3Translational Cardiovascular Research Center, College of Medicine-Phoenix, University of Arizona, Phoenix, AZ 85004, USA; 4BIO5 Institute, University of Arizona, Tucson, AZ 85721, USA; 5Sarver Heart Center, University of Arizona, Tucson, AZ 85721, USA

**Keywords:** pulmonary vasculature, vascular biology, BMP, angiogenesis, lung

## Abstract

Transmembrane protein 100 (TMEM100) is a crucial factor in the development and maintenance of the vascular system. The protein is involved in several processes such as angiogenesis, vascular morphogenesis, and integrity. Furthermore, TMEM100 is a downstream target of the BMP9/10 and BMPR2/ALK1 signaling pathways, which are key regulators of vascular development. Our recent studies have shown that TMEM100 is a lung endothelium enriched gene and plays a significant role in lung vascular repair and regeneration. The importance of TMEM100 in endothelial cells’ regeneration was demonstrated when Tmem100 was specifically deleted in endothelial cells, causing an impairment in their regenerative ability. However, the role of TMEM100 in various conditions and diseases is still largely unknown, making it an interesting area of research. This review summarizes the current knowledge of TMEM100, including its expression pattern, function, molecular signaling, and clinical implications, which could be valuable in the development of novel therapies for the treatment of cardiovascular and pulmonary diseases.

## 1. Introduction

Endothelial cells (ECs) are a vital component of the inner wall of blood vessels and play a crucial role in maintaining the health and stability of tissues. ECs regulate blood flow, exchange oxygen and nutrients, and preserve the integrity of blood vessels, ensuring proper tissue function [1]. Although the heterogeneity of ECs across different organs has been well documented [2], advances in single-cell RNA-sequencing (scRNA-seq) have allowed for a better understanding of the heterogeneity of ECs across different organs, revealing the unique features of ECs in each organ [3,4,5]. Compared to other organs, the lung is of particular interest due to its unique morphological composition, consisting of a thin layer of capillary ECs that enable efficient gas exchange [6]. Our lab’s long-term research interest is on lung vascular biology and pulmonary vascular diseases [7,8]. Recent findings and publications have uncovered a previously unknown transmembrane protein 100 (TMEM100) that is enriched in lung ECs. TMEM100 is abundantly expressed in the lung vasculature compared to other organs. The genetic deletion of Tmem100 and constitutive deletion of Tmem100 using Tek-Cre (ECs and hematopoietic stem cells-specific) in mice produce embryonic lethality, suggesting the critical role of TMEM100 in vascular development [9,10]. Additionally, TMEM100 is downregulated in various kinds of cancers including lung cancer and other lung diseases, and contributes to disease progression. TMEM100 has also been demonstrated to contribute neurological disease, pain management, and the regulation of inflammatory process.

Given the potential importance of TMEM100 in both physiological and pathological conditions, it is essential to understand its role and underlying mechanisms. In this review, we aim to summarize the current knowledge of TMEM100, including its expression pattern, function, upstream and downstream signaling of TMEM100, and its potential significance and perspective in clinical applications. We used TMEM100 as the key word to search the PubMed database. To the best of our knowledge, this is the first review article to focus on and summarize the role and mechanisms of TMEM100 in the pathophysiological processes. We believe our review article will benefit the researchers in the field. 

## 2. TMEM100 and Its Expression Pattern

TMEM100 is a unique protein that possesses two putative transmembrane domains and an extracellular fragment, with its N and C terminals located intracellularly. Human TMEM100 is a small protein with 134-amino acid. AlphaFold prediction [11] shows that the YSFNSHGSI fragment might be the extracellular domains (Figure 1A). It is structurally unique from any known family of proteins. TMEM100 is well conserved across different species including rodent and chicken (Figure 1B). Interestingly, the subcellular localization of TMEM100 has been a subject of debate among researchers. Some studies have reported that TMEM100 is located in the plasma membrane, while others have found it to be enriched in the endoplasmic reticulum (ER) membrane fraction. Previous literature has reported the localization of TMEM100 to be in the plasma membrane in the HEK293 cell under TMEM100 overexpression using anti-TMEM100 antibodies [12]. However, Somekawa et al. found that endogenous TMEM100 was located and enriched in the ER membrane fraction through western blot analysis by performing subcellular fractionation and preparing ER microsome fractions in human umbilical vascular endothelial cells (HUVECs). [10]. They also demonstrated that the forced overexpression of TMEM100-FLAG was co-localized with an ER protein GRP78/HSPA5 but no other organelle markers in sub-confluent HUVECs [10]. Our unpublished data showed that TMEM100 locates both in the cell membrane and ERs using the TMEM100-Tdtomato fused protein and immunostaining against TMEM100 or FLAG antibodies in human primary microvascular ECs (hPMVECs). Thus, the sub-cellular localization of TMEM100 expression in the human and mouse pulmonary vascular ECs in vitro and in vivo remains uncertain. These conflicting results highlight the need for further research to fully understand the subcellular localization of TMEM100 in different cell types in the future.

Tmem100 mRNA is expressed in mouse embryos from embryonic day 8.5 (E8.5) and is found to have elevated levels at later stages, as analyzed by Northern blot analysis. Additionally, in adult mice, Tmem100 was found to be most abundantly expressed in the lung in contrast to lower expression levels in the brain, heart, and muscle using northern blot analysis [10]. Recent scRNA-seq analysis of the murine ECs from multiple organs also showed that TMEM100 is one of the top enriched genes in the lung vasculature compared to other organ vasculatures [13,14]. Our recent studies using immunostaining against TMEM100 in different human organs showed that TMEM100 is highly expressed in the human lung ECs but not in any other organ (heart, liver, kidney) ECs [14], suggesting that TMEM100 might play an important role in lung vasculature hemostasis. Moreover, several groups have generated Tmem100 reporter mouse lines [9,10,14,15,16] (Table 1). Moon et al. and Somekawa et al. showed that Tmem100 reporter expression was detected in the major arteries in E10.5 and E11.5 embryos [9,10]. Kinugasa-Katayama et al. showed that Tmem100-BAC-EGFP reporter expression is the highest expressed in the lung, and is also expressed in the bone and esophagus [15]. In the embryonic lung, Tmem100-BAC-EGFP reporter expression was observed in the vascular cells and mesenchymal cells. In the embryonic heart, reporter expression was evident in the coronary arteries, endocardium, and tricuspid valve. Moon et al. showed that Tmem100 expression was also found in the mammary glands, notochord, and ventral regions of the neural tube [9]. Vesprey et al. employed a Tmem100-creERT2; Ai14 reporter mouse line and found that postnatal Tmem100-creERT2 expression marked endothelial cells, PaS cells, osteoblasts, and proliferating zone chondrocytes [16]. Through colocalization with a pan-neuronal marker, PGP9.5, Eisenman et al. found the localization of TMEM100 immunoreactivity in the mice and human enteric nervous system and the mice’s central and peripheral nervous system in addition to arterial ECs [12]. Our recent studies using Tmem100-CreERT2; Ai6 mice demonstrated that Tmem100 (ZsGreen) is mainly expressed in the lung ECs and the right atrium in adult mice. Further examination of Tmem100 (ZsGreen) expression in the lung showed that Tmem100 is highly expressed in 92% of the capillary ECs, 72% of arterial ECs, and 46% of venous ECs. Tmem100 is also expressed in some pericytes, but not smooth muscle cells [14]. A summary of the Tmem100 reporter mouse models can be found in Table 1. Some limitations with these Tmem100-related mouse models were that they were only explored in certain organs or stages for Tmem100 expression in these animals. It is still possible that Tmem100 is expressed or diminished in a spatiotemporal manner within certain cells.

## 3. Role of TMEM100

As a downstream target of the Bone Morphogenetic Protein (BMP) 9/10 and Activin-Like Kinase Receptor Type I (ALK1) signaling pathways, TMEM100 plays an important role in angiogenesis, vascular morphogenesis, integrity, and cardiovascular development. BMP9/10 acts mainly via an EC-specific ALK1 receptor and promotes arterial endothelial maturation and quiescence. The disruption of ALK1, or a type III coreceptor endoglin (ENG), caused embryonic lethality with defects in arterial endothelium differentiation and vascular morphogenesis [17,18]. In Tmem100-null mouse embryos, impaired retinal angiogenesis and abnormal vasculature branching and density were observed via the immunolabeling of blood vessels using Fluorescein isothiocyanate (FITC)-conjugated isolectin B4 [19]. Tmem100 deficiency in adult mice also increased vascular permeability and hemorrhage in various organs. Moon et al. reported AV shunting in the liver and intestine in TMEM100-deficient adult mice via the injection of a latex dye in the left ventricle of the heart, which was milder compared to those in Alk1-deficient mice [20]. In the lungs of Tmem100-deficient mice, the researchers also found that there was leakage of the latex dye, indicating a compromise in the vascular integrity [20]. Further analysis by Gene Set Enrichment Analysis (GSEA) revealed that genes encoding cell adhesion and extracellular matrix (ECM) proteins were most significantly affected in the lungs of Tmem100-knockout mice, highlighting the critical role of TMEM100 in regulating these processes [19]. Our recent studies showed that the overexpression of TMEM100 did not affect the EC cellular barrier integrity under the Thrombin challenge [14], suggesting that the regulation of vascular permeability by Tmem100 in vivo may be due to the indirect regulation of the ECM but not the EC junction. Furthermore, Somekawa et al. showed that Tmem100 null embryos showed fetal defects of arterial endothelium differentiation and severe vascular dysmorphogenesis and cardiac enlargement at E9.5, and massive pericardial effusion and growth retardation at E10.5 [10]. These phenotypes are identical to the Acvrl1 deficiency mice [18]. Our recent studies demonstrated that the overexpression of TMEM100 in human pulmonary microvascular ECs promoted EC proliferation and tube formation in vitro. The loss of Tmem100 in ECs in mice exhibited impaired lung EC regeneration following inflammatory lung injury [14]. Taken together, these studies suggest that TMEM100 play an important role in EC homeostasis and regeneration (Figure 2).

Moreover, TMEM100 was reported to play an important role in lymphangiogenesis in a study by Hye et al. [20]. In this study, the authors generated a Tmem100-overexpression mouse line and Tmem100-null mice to overexpress and knockout Tmem100, respectively. In Tmem100-null mice, the researchers found that TMEM100 deficiency led to blood-filled lymphatic vessels and the enlargement of lymphatic sacs and vessels, suggesting a misconnection between the blood and lymphatic vasculature [20]. Additionally, by performing immunofluorescence analyses on the cardinal vein of embryos, the researchers reported abnormal lymphatic endothelial cell (LEC) specification in Tmem100-knockout mice, which could have led to a misconnection between the blood and lymphatic vasculature [20]. On the other hand, Tmem100 overexpression led to severe subcutaneous edemas with reduced lymphatic drainage and a reduced number and size of lymphatic vessels and sacs [20]. Tmem100 overexpression also led to fewer LEC progenitors and inhibited early LEC specification, indicating that Tmem100 is essential to LEC specification and lymphangiogenesis [20]. To better understand the molecular mechanisms behind TMEM100′s role in lymphangiogenesis, the researchers investigated the activity of NOTCH, which was known to be a negative regulator of LEC specification. Immunofluorescence analyses revealed that NOTCH was downregulated in the cardinal vein of Tmem100-knockout mice and upregulated in Tmem100-overexpressed mice, suggesting that NOTCH signaling plays an essential role in the Tmem100 pathway [20].

Recently, Karolak et al. analyzed the molecular change of lethal lung developmental disorders (LLDDs), including alveolar capillary dysplasia with misalignment of pulmonary veins (ACDMPV), acinar dysplasia (AcDys), congenital alveolar dysplasia (CAD), and primary pulmonary hypoplasia, which involve the loss-of-function of *FOXF1*, *TBX4,* or *FGF10.* Through transcriptome analysis, they showed that TMEM100 is one of the six significant deregulated genes in the lungs of patients with either TBX4 or FGF10 variants. They confirmed a marked decrease in TMEM100 staining within the endothelium of arteries and capillaries in all cases of LLDDs compared with controls [21,22]. These data suggest that TMEM100 deficiency plays an important role in LLDDS pathogenesis. Moreover, our recent study found that TMEM100 is downregulated in a subpopulation of ECs and the whole lung ECs of PH mice (*Egln1^Tie2Cre^*), a mouse model with severe PH with progressive obliterative vascular remodeling including vascular occlusion and plexiform-like lesion and right heart failure [7,23,24]. Tmem100 was also downregulated in the lungs of mice with PH induced by the overexpression of dominant-negative Bmpr2 [25]. Given the important role of BMP signaling in pulmonary arterial hypertension (PAH), these data suggest that TMEM100 might contribute to the pathogenesis of PAH.

Interestingly, TMEM100 also plays a role as a tumor-suppressor gene [26]. In a study by Frullanti et al., the authors first isolated noninvolved lung tissue and lung adenocarcinoma tissue and extracted RNA from the samples to perform a gene expression analysis [27]. In functional studies in vitro, the authors found that TMEM100 inhibited colony formation and tumor growth when overexpressed in lung cancer cell lines [27]. Furthermore, from gene expression analysis on the samples, the authors found that TMEM100 transcripts were reduced in less-differentiated cells, such as invasive lung adenocarcinomas, suggesting the tumor-suppressor capabilities of TMEM100 [27].

TMEM100 has also been involved in the metastasis and proliferation of cancer cell lines, and mediates inflammatory pathways and the secretion of inflammatory cytokines. In a study by Pan et al., the authors first transfected LX-2 cells, a human hepatic stellate cell line, with pEGFP-C2-TMEM100 and TMEM100 siRNA to overexpress and silence TMEM100, respectively, and found that overexpressing or silencing TMEM100 led to a significant inhibition or promotion of proliferation, respectively [28]. Treatment with TNF-α in the cells led to a downregulation of TMEM100 in addition to the upregulation of IL-1β and IL-6 by western blot analysis; however, the subsequent upregulation of TMEM100 led to a reduction of IL-1beta and IL-6 levels, indicating the regulatory effect of TMEM100 in cytokine secretion [28]. TMEM100 overexpression significantly reduced phosphorylated ERK and JNK in TNF-alpha treated LX-2 cells, suggesting that TMEM100 may also be an essential mediator in the classical MAPK [28]. In a study by Li et al., the researchers first identified the poor expression of TMEM100 in colorectal cancer cells (CRC) by screening differentially expressed mRNAs and qRT-PCR [29]. Through (3-(4,5-Dimethylthiazol-2-yl)-2,5-Diphenyltetrazolium Bromide (MTT) analysis, the researchers also found that CRC proliferation and colony formation was noticeably enhanced after TMEM100 silencing but significantly weakened after TMEM100 overexpression [29].

Additionally, scratch healing and Transwell assays showed that CRC migratory and invasive ability was markedly increased after TMEM100 silencing and was inhibited upon TMEM100 overexpression, suggesting that TMEM100 plays a vital role in restraining CRC proliferation, migration, and invasion [29]. Further analysis by the researchers through assays such as western blot and GSEA revealed that TMEM100 regulated the TGF-beta pathway, and this regulation suppressed CRC cell growth by repressing TGF-β signaling pathway activation [29]. In another study involving non-small-cell lung carcinoma (NSCLC), Han et al. reported that TMEM100 upregulation led to the inhibition of cell proliferation, migration, and invasion in NSCLC [30]. The researchers found that overexpressing TMEM100 in NSCLC inhibited cell proliferation and metastasis, conversely knocking down TMEM100 promoted cell proliferation ability in NSCLC [30]. Finally, in a study by Wang et al., the researchers showed that TMEM100 was a factor in inhibiting the epithelial-mesenchymal transition (EMT) in NSCLC [31]. By overexpressing TMEM100 in NSCLC cells, the researchers found that EMT expression markers were impaired, the morphology of NSCLC cells changed from mesenchymal back to epithelial, and there was impairment in the proliferative ability of NSCLC cells [31]. Moreover, TMEM100 was markedly downregulated in pancreatic cancer (PCa) tissues. The overexpression of TMEM100 attenuates the proliferation, migration, invasion and EMT in DU145 cells, a PCa cell line [32]. Similarly, TMEM100 was shown to decrease in the prostate cancer patients, and low TMEM100 expression was associated with advanced tumor stage and metastasis. The overexpression of TMEM100 suppressed prostate cancer cell progression by inhibiting the FAK/PI3K/AKT signaling pathway [33]. Recent studies also showed that TMEM100 was identified as a diagnostic gene for gastric cancer via re-analyzing the existing gastric cancer datasets. TMEM100 was significantly positively correlated with memory CD4+ T cell and activated mast cells, and negatively correlated with M0 Macrophage and activated memory CD4+ T cells. The expression of TMEM100 effectively predicts overall survival, first progression and post-progression survival in gastric cancer patients [34]. Indeed, TMEM100 expression was significantly downregulated in gastric cancer patients. The overexpression of TMEM100 inhibited the migration and invasion without affecting tumor growth. Moreover, the overexpression of TMEM100 sensitized gastric cancer cells to chemotherapeutic drugs [35]. Park et al. also identified the TMEM100 gene to be a biomarker that can best distinguish the invasive and mitotic subtypes of glioblastoma, as the expression levels of TMEM100 were significantly correlated with mitotic subtypes [36].

TMEM100 is expressed in both neurons and perineuronal glial cells in the rat dorsal root ganglion (DRG). The TMEM100 protein is significantly increased in the lumbar DRGs in the complete Freund adjuvant inflammatory pain. In contrast, peripheral nerve injury induced by spinal nerve ligation diminishes TMEM100 expression in the DRG [37]. Furthermore, in other studies, TMEM100 has also been shown to modulate pain reception. In a study by Weng et al., researchers showed that TMEM100 modulates the pain response by regulating the calcium-dependent TRPA-V1 complex in the nociceptive pathway by weakening the association between TRPA and TRPV1 [38]. The researchers generated conditional Tmem100 knockout mice through selective breeding and found through behavioral tests that Tmem100 deficiency weakened TRPA1-associated nociception and inflammatory hyperalgesia in mice [38]. Interestingly, TRPV1-associated nociception and inflammatory hyperalgesia were not perturbed in the mutant mice, indicating that TMEM100 did not have a modulatory effect on TRPV1 activity [38]. Immunoprecipitation experiments and Glutathione S-transferase (GST)-pull down studies revealed that TMEM100 binds both TRPA1 and TRPV1 in the complex and can physically interact with them, removing TRPV1′s inhibition on TRPA1 [38]. Recent studies showed that TMEM100+ DRG neurons account for most of the activated neurons in an aceton-ether-water (AEW)-induced dry skin itch model. TMEM100+ DRG neurons are colocalized with TRPA1 and the chloroquine-related Mrgpr itch receptor family demonstrated by single-cell RNA sequencing analysis and immunostaining. Specific DRG TMEM100 deficiency reduced AEW-induced itch and TRPA1′s functional change, indicating the therapeutic potential of TMEM100 suppression in patients with dry skin-induced itch [39].

## 4. Molecular Signaling Affected by TMEM100

TMEM100 regulates MFAP4, a gene involved in cell adhesion and ECM proteins in the lung that localizes to elastic fibrils of arterioles and alveolar walls [19]. After performing gene expression analysis in Tmem100-knockout embryos, the authors found that MFAP4 was the most significantly downregulated and had a lower number of elastin-positive vessels and elastin density [19]. Additionally, TMEM100-deficiency led to an impairment in the formation of endothelial and smooth muscle cell layers in the lung in Tmem100-knockout lungs [19]. Moreover, TMEM100 regulates NOTCH, a negative regulator of the specification of lymphatic endothelial cells [20,40]. In Tmem100-deficient embryos, there was a suppression of NOTCH signaling in arterial ECs and the endocardium and impairments in lymphedema and enlarged lymphatic vessels [20]. Conversely, Tmem100-overexpression enhanced NOTCH signaling and led to small and disorganized lymphatic vessels [20], suggesting the positive regulation of NOTCH signaling by TMEM100 (Figure 2).

Furthermore, TMEM100 is involved in cell proliferation and apoptosis in mouse embryonic kidney-derived cells by negatively regulating BMPR-II and BMP7 [41]. In Tmem100 knockout cells, TMEM100 deficiency increased BMPR-II and BMP7 expression, cell proliferation, and the apoptosis of MK3 cells, an early metanephric mesenchyme cell line [41]. Interestingly, BMP7 plays a more important role than BMPR-II in the TMEM100-mediated cell proliferation of MK3 cells, as shown by the decrease of cell proliferation in TMEM100-depleted cells caused by a downregulation of BMP7, not BMPR-II [41]. Our recent study demonstrated that the overexpression of TMEM100 in hPMVECs enhanced cell proliferation, tube formation, the enrichment of the cell cycle pathway assessed by KEGG, and the upregulation of many genes related to cell proliferation including FOXM1, E2F1, and PLK1, etc., [14].

Furthermore, TMEM100 is also involved in inflammatory pathways. By regulating the TGF-β pathway, TMEM100 suppresses CRC growth and metastasis by impairing TGF-β’s ability to bind with its transmembrane receptors [29]. TMEM100 overexpression in TNF-α-treated LX-2 cells also inhibited IL-1β and IL-6, showing that TMEM100 is critical for cytokine secretion [28]. TMEM100 is also involved in the MAPK signaling pathway; TMEM100 overexpression reduced the expression of phosphorylated ERK and phosphorylated JNK in TNF-α stimulated LX-2 cells [28]. In other studies, TMEM100 was involved in the Wnt/β-Catenin pathway, through the blockage of nuclear translocation of β-Catenin, and the TNF signaling pathways to impair the metastasis and proliferation of NSCLC [30,31]. The overexpression of TMEM100 shortens the half-life of hypoxia-inducible factor-1α (HIF-1α) via ubiquitination and the proteasome pathway and inhibits colorectal cancer cells’ migration and angiogenesis potential [42]. It will be interesting to study the role of TMEM100 in the lung vasculature given the high abundance and uniqueness of HIF-2α in the lung vasculature system.

## 5. Regulation of TMEM100 Expression

TMEM100 plays a crucial role in various processes, including angiogenesis, the determination of arterial cell fate, the maintenance of vascular integrity, and its potential involvement in the pathogenesis of diseases such as PAH. In a study by Somekawa et al., the researchers first found that TMEM100 mRNA and protein expression was induced by BMP9 or BMP10 treatment in HUVECs. These effects were significantly inhibited by the knockdown of ALK1, BMPR2 and SMAD4. They then confirmed the downregulation of TMEM100 mRNA in the arterial endothelium in Alk1-null mouse embryos through qRT-PCR [10]. These data demonstrate that TMEM100 is a downstream target of BMP9/10/ALK1/BMPR2 signaling. In addition, Chen et al. showed that the expression of TMEM100 was decreased after exposure to hyperoxia in rat pups. They also showed that BMP9 treatment induced TMEM100 mRNA expression in pulmonary arterial ECs [43]. The downregulation of TMEM100 was also evident in the placentas of the paternally expressed 11/Retrotransposon-like 1 (Peg11/Rtl1) KO mice, which has mid to late fetal lethality or late fetal growth retardation associated with frequent neonatal lethality [44]. Importantly, TMEM100 is also a potential downstream target of FOXF1 using ChIP-seq analysis in mice with *Foxf1* overexpression, suggesting that FOXF1 abnormalities may trigger the downregulation of this gene in ACDMPV patients [22]. Recent studies showed that TMEM100 is the downstream target of MEF2C in human induced pluripotent stem cell (iPSC)-derived ECs. They showed that the MEF2C binding peaks on the promoter region of TMEM100 was present in the control hiPSC-ECs, but absent in the MEF2C knockdown hiPSC-ECs by ChIP-seq analysis, demonstrating that MEF2C directly binds to the promoter region of TMEM100 [45]. Other studies showed that global gene expression profiling shows that the knockdown of ANRIL (DQ485454), a gene involved in atherosclerosis, at 9p21.3 genome-wide association studies (GWAS) cardiovascular disease (CAD) locus upregulates TMEM100 [46].

Our unpublished data also showed that pro-angiogenesis factor VEGF-A and pro-inflammatory factor TNF-α induced TMEM100 expression, whereas hypoxia treatment inhibited TMEM100 expression in primary rat pulmonary vascular ECs in vitro. Moreover, we found that TMEM100 expression was dramatically reduced in the inflammatory-induced injured lungs. However, we did not observe a significant change of TMEM100 expression in the lungs of hypoxia treated mice.

TMEM100 expression is downregulated in multiple cancer tissue and cell lines, including lung adenocarcinoma (LUAD) and lung squamous cell carcinoma (LUSC), colorectal cancer, gastric cancer, and PCa [26,29,32,35,47]. Liu et al. demonstrated that GATA binding 5 (GATA5), a transcription factor of GATA protein family, binds to the TMEM100 promoter and transcriptionally activates TMEM100 and suppresses PCa progression [32]. Furthermore, Wang et al. found that there was an obvious H3K27ac modification in the promoter sequence in TMEM100 and a significantly negative correlation between TMEM100 and histone deacetylase 6 (HDAC6) expression in TCGA-LUAD and LUSC patients [31]. HDAC6 modulates TMEM100 expression by repressing it in NSCLC, leading to the increased migration and invasiveness of NSCLC [31]. TMEM100 is also directly regulated by miRNA-421 in LUAD and miRNA-106 in NSCLC [48,49]. Moreover, TMEM100 was upregulated by enhanced circ_0000567 in LUAD cells, and the expression of TMEM100 was inversely proportional to miR-421 in LUAD cells [48].

Interestingly, ATF6a, a member of the Unfolded Protein Response (UPR) transcription factor family responsible for sensing ER stress, was shown to affect the expression of TMEM100 only under the stimulation of thapsigargin, an inducer of ER stress [50]. Ionomycin, an ionophore, is also a reagent that upregulates TMEM100 expression [50]. Treatment with these reagents increased TMEM100 mRNA levels in HEK293T cells, and treatment with EGTA, a calcium chelator, abolished this increase. These results suggest that cytosolic calcium ions are essential for regulating TMEM100 expression [50]. TMEM100 is also regulated by inflammatory signaling. Pan et al. found that TNF-α downregulated TMEM100 at the protein and mRNA level in the LX-2 cell line [28].

## 6. Discussion and Therapeutic Perspective

TMEM100 is a two-transmembrane protein that is known to play various roles in different physiological processes in the body. Studies have shown that it is primarily concentrated in the lung vascular system during the adult stage and plays a significant role in angiogenesis, vascular permeability, and lymphatic system development. Additionally, TMEM100 has been shown to affect inflammatory pathways and act as a tumor suppressor in specific cells, such as NSCLC, as depicted in Figure 2.

Given these diverse roles, it is essential to further investigate the specific roles that TMEM100 plays in the development of pathological conditions, such as PAH and kidney failure, as well as the progression of various cancers. Future studies should focus on identifying the molecular relationships between TMEM100 and its upstream and downstream regulators and how these relationships affect vascular development and the pathogenesis of pulmonary diseases.

Studies have found that TMEM100 expression is downregulated in the lungs during inflammatory lung injury and pulmonary hypertension, and that Tmem100 deficiency impairs lung vascular regeneration after injury. These findings suggest that the activation of Tmem100 signaling may represent a novel strategy for lung vascular repair and regeneration. Further research is necessary to elucidate the precise mechanisms by which TMEM100 functions and to explore the potential therapeutic implications of targeting this protein for the treatment of various diseases.

In addition to its various functions in the body, TMEM100 also possesses a unique structural feature: the extracellular domain of TMEM100 is structurally distinct from any known family of proteins, according to protein sequence BLAST data. Moreover, this extracellular domain is highly conserved across different species, including rodents and chickens. These characteristics make it a promising target for the development of specific peptides or antibodies that can bind to TMEM100’s extracellular domain. TMEM100 is expressed most abundantly in the lungs compared to all other organs. In cases where there is cell death due to the progression of lung diseases such as lung cancer or PAH, TMEM100 could potentially be released into the circulation. Therefore, there is a possibility that TMEM100 could serve as a biomarker for the early detection of various lung diseases, including lung cancer and PAH.

Conjugating such peptides or antibodies with pharmaceutical agents, such as therapeutic drugs or imaging probes, could potentially enhance tissue specificity and lung efficacy while decreasing systemic side effects. This targeted delivery approach could revolutionize the field of lung-specific pharmaceuticals and offer a new avenue for the treatment of various pulmonary diseases. The development of peptides or antibodies that selectively target TMEM100 may also offer diagnostic and therapeutic opportunities for the early detection and treatment of lung cancer given TMEM100’s role as a tumor suppressor in specific cells. Further research is needed to explore the potential clinical applications of targeting TMEM100, including the development of TMEM100-specific drugs, peptides, and antibodies.

## Figures and Tables

**Figure 1 biomedicines-11-00937-f001:**
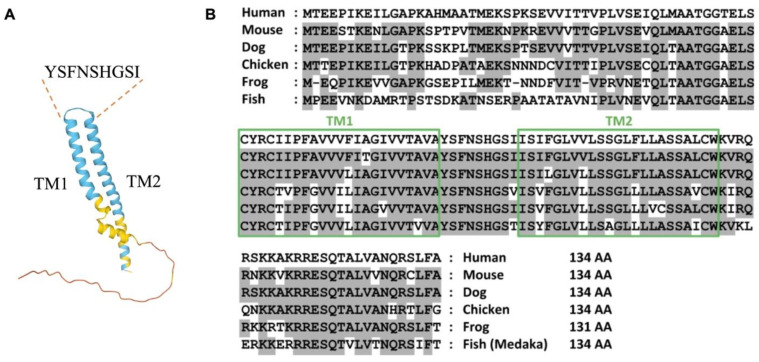
TMEM100. (**A**) The diagram of AlphaFold prediction of TMEM100. TM: transmembrane domain. YSFNSHGSI indicates the putative extracellular domain of TMEM100. (**B**) Amino acid sequences of TMEM100 from multiple species. Boxes represent two putative transmembrane (TM) domains. Residues conserved with the human sequence are shaded in gray. Reproduction from ref. [10].

**Figure 2 biomedicines-11-00937-f002:**
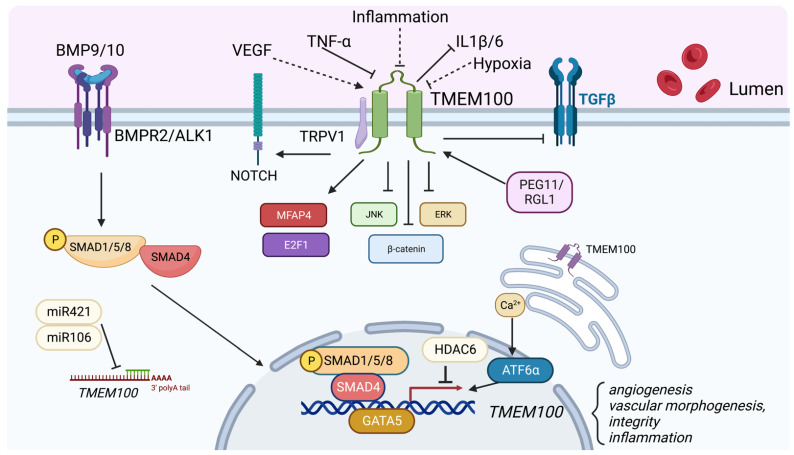
Role of TMEM100. TMEM100 is a two transmembrane protein located in the plasma membrane and endoplasmic reticulum membrane. TMEM100 is a BMP9/10-BMPR2/ALK1-pSMAD1/5/8 signaling downstream target. Various factors such as VEGF, PEG11/RGL1, GATA5, Calcium signaling and ATF6α upregulate TMEM100 expression. Inflammatory cytokines such as TNF-α, miR421/miR106, and HDAC6 suppress TMEM100 expression. TMEM100 has been demonstrated to positively regulate NOTCH, MFAP4, and E2F1 signaling, and inhibit JNK/ERK, β-catenin, TGF-β signaling. Our unpublished data showed that hypoxia and inflammatory injury reduce TMEM100 expression in the lung ECs, whereas VEGF upregulates TMEM100 expression. (indicated by dotted lines).

**Table 1 biomedicines-11-00937-t001:** Summary of published Tmem100 reporter mouse models.

Mouse Line	Reporter Gene	Expression Pattern		Reference
		Embryonic	Adult	
Tmem100-SIBN	LacZ/X-gal	E12.5: predominantly at arteries, heart, mammary glands, notochord, neural tube.	N/A	[9]
Tmem100 null/lacZ	LacZ/X-gal	E9.5: arteries and endocardium. E10.5: limb bud, hindgut, somite.	P30, vascular EC, alveolar cells	[10]
Tmem100-BAC-EGFP	EGFP	E9.5 arteries, endocardium. E15.5, lung mesenchymal cell, coronary arteries, endocardium, cardiac valve.	N/A	[15]
Tmem100.creERT2 (BAC) JAX: 014159	Ai14:Tdtomato	N/A	P13, P18, P120, P180, osteocytes in the diaphyseal cortex and chondrocytes in the growth plate.	[16]
Tmem100.creERT2 (BAC) JAX: 014159	Ai6: ZsGreen	N/A	Adult, lung endothelium, including capillaries, arteries, vein. Right atrium, and cardiac valve.	[14]

## Data Availability

Not applicable.

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
