# Peer review of "The Role of a Lung Vascular Endothelium Enriched Gene TMEM100"

_biomedicines, 2023, doi:10.3390/biomedicines11030937_

Round 1

Reviewer 1 Report

In this review, the authors have highlighted the signaling role of TMEM-100 in endothelial cell repair, regeneration, and maintaining vascular integrity. This review will certainly help the scientific community more strategically to resolve the endothelial cells injury and organ dysfunction.

The manuscript is well written, it should be accepted since it will add knowledge for the scientific community. However, it needs some minor revision before acceptance.

1.     The authors have mentioned many places citing their unpublished data. It would be useful to readers if the authors could highlight these in a graphical (figure form).

2.     In section 5 (Regulation of TMEM-100 Expression) second para, it seems the 1st sentence is incomplete and/or authors can re-write it.

3.     Authors must thoroughly check typing errors in literature citing throughout manuscript text.

Author Response

Review 1:

Comments: In this review, the authors have highlighted the signaling role of TMEM-100 in endothelial cell repair, regeneration, and maintaining vascular integrity. This review will certainly help the scientific community more strategically to resolve the endothelial cells injury and organ dysfunction.

The manuscript is well written, it should be accepted since it will add knowledge for the scientific community. However, it needs some minor revision before acceptance.

Response: Thank you for your positive comments.

Critique 1. The authors have mentioned many places citing their unpublished data. It would be useful to readers if the authors could highlight these in a graphical (figure form).

Response: We incorporated our unpublished data in the Figure 2 and labeled with dotted lines.

Critique 2. In section 5 (Regulation of TMEM-100 Expression) second para, it seems the 1st sentence is incomplete and/or authors can re-write it.

Response: The sentence was rewritten. Thanks for the suggestion.

Critique 3. Authors must thoroughly check typing errors in literature citing throughout manuscript text. 

Response: The language was edited by two native English speakers. 

Reviewer 2 Report

Title: The Role of a Lung Vascular Endothelium Enriched Gene 2
TMEM100

 Authors: Jiakai Pan, Bin Liu and Zhiyu Dai

Summary:

The authors provide an overview of the current knowledge of transmembrane protein 100 (TMEM100) and its potential importance in the field of pulmonary vascular biology.

This manuscript is an interesting piece of work, however, the missing information must be filled in, and if it is not available, the limitations of the available results must be discussed in detail.

Several major points are listed below:

1: The summary is too short and lacks content and should be rewritten as it seems unstructured.

2: The novelty of the article should be clearly emphasized throughout.

3: The introduction is too short and uniform. Much more should be written about TMEM100.

4: The novelty of the article should be clearly emphasized.

5: The search strategy used for the literature review should be stated.

6: The limitations associated with the use of TMEM100 should be discussed.

7: I miss a separate overall discussion and conclusion here.

8: All abbreviations should be defined at their first mention and used thereafter.

9: Please use consistent abbreviations throughout the manuscript.

10: The paper contains many complicated abbreviations. A list of the most important abbreviations would be helpful to the reader.

11: The English language in general needs improvement. The entire text needs to be revised by a native English speaker.

Author Response

Review 2:

Comments: The authors provide an overview of the current knowledge of transmembrane protein 100 (TMEM100) and its potential importance in the field of pulmonary vascular biology.

This manuscript is an interesting piece of work, however, the missing information must be filled in, and if it is not available, the limitations of the available results must be discussed in detail.

Response: Thank you for your comments. We made significant changes to the manuscripts and responded to each of the comments.

Critique 1: The summary is too short and lacks content and should be rewritten as it seems unstructured. 

Response: We rewrite our abstract based on reviewer’s suggestion.

Critique 2: The novelty of the article should be clearly emphasized throughout.

Response: The role and mechanism studies of TMEM100 were not well documented. Our review article is the first to summarize the current knowledge of TMEM100. It was emphasized in the Introduction.

Critique 3: The introduction is too short and uniform. Much more should be written about TMEM100.

Response: We expanded the Introduction section as suggested and included the critical role of TMEM100 in pathophysiological conditions.

Critique 4: The novelty of the article should be clearly emphasized. 

Response: The role and mechanism studies of TMEM100 were not well documented. Our review article is the first to summarize the current knowledge of TMEM100. It was emphasized in the Introduction.

Critique 5: The search strategy used for the literature review should be stated.

Response: The search strategy was stated in the introduction.

Critique 6: The limitations associated with the use of TMEM100 should be discussed.

Response: The limitations associated with the use of TMEM100 animal models was discussed in section 2.

Critique 7: I miss a separate overall discussion and conclusion here.

Response: The discussion and conclusion were embedded in the section 6.

Critique 8: All abbreviations should be defined at their first mention and used thereafter.

Response: The abbreviations were defined at their first mention and used thereafter according to reviewer’s suggestion.

Critique 9: Please use consistent abbreviations throughout the manuscript.

Response: The abbreviations was consistent now after carefully checked.

Critique 10: The paper contains many complicated abbreviations. A list of the most important abbreviations would be helpful to the reader.

Response: A list of abbreviation was included.

Critique 11: The English language in general needs improvement. The entire text needs to be revised by a native English speaker.

Response: The language was edited by two native English speakers. 

Reviewer 3 Report

There is a possible role for this protein as a clinical marker for some lung diseases?  May it be found by histological techniques?

Author Response

Reviewer 3:

Critique 1: There is a possible role for this protein as a clinical marker for some lung diseases?  May it be found by histological techniques?

Response: Thank you for your suggestion. TMEM100 is expressed most abundantly in the lungs compared to all other organs. In cases where there is cell death due to the progression of lung diseases such as lung cancer or PAH, TMEM100 could potentially be released into the circulation. Therefore, there is a possibility that TMEM100 could serve as a biomarker for the early detection of various lung diseases, including lung cancer and PAH. TMEM100 could be identified by immunostaining and RNASCOPE.

Round 2

Reviewer 2 Report

The authors have addressed all points of potential criticism and each suggestion made by the reviewer adequately and in detail.